# Comparing Multiple Locus Variable-Number Tandem Repeat Analyses with Whole-Genome Sequencing as Typing Method for *Salmonella* Enteritidis Surveillance in The Netherlands, January 2019 to March 2020

Roan Pijnacker,[a] Maaike van den Beld,[a] Kim van der Zwaluw,[a] Anjo Verbruggen,[a] Claudia Coipan,[a] Alejandra Hernandez Segura,[a] Lapo Mughini-Gras,[a,b] Eelco Franz,[c] Thijs Bosch[c]

[a]Centre for Infectious Disease Control, National Institute for Public Health and the Environment (RIVM), Bilthoven, the Netherlands
[b]Institute for Risk Assessment Sciences, Utrecht University, Utrecht, the Netherlands
[c]Centre for Infectious Disease Control, National Institute for Public Health and the Environment (RIVM), Bilthoven, the Netherlands

**ABSTRACT** In the Netherlands, whole-genome sequencing (WGS) was implemented as routine typing tool for *Salmonella* Enteritidis isolates in 2019. Multiple locus variable-number tandem repeat analyses (MLVA) was performed in parallel. The objective was to determine the concordance of MLVA and WGS as typing methods for *S.* Enteritidis isolates. We included *S.* Enteritidis isolates from patients that were subtyped using MLVA and WGS-based core-genome Multilocus Sequence Typing (cgMLST) as part of the national laboratory surveillance of *Salmonella* during January 2019 to March 2020. The concordance of clustering based on MLVA and cgMLST, with a distance of ≤5 alleles, was assessed using the Fowlkes-Mallows (FM) index, and their discriminatory power using Simpson's diversity index. Of 439 isolates in total, 404 (92%) were typed as 32 clusters based on MLVA, with a median size of 4 isolates (range:2 to 141 isolates). Based on cgMLST, 313 (71%) isolates were typed as 48 clusters, with a median size of 3 isolates (range:2 to 39 isolates). The FM index was 0.34 on a scale from 0 to 1, where a higher value indicates greater similarity between the typing methods. The Simpson's diversity index of MLVA and cgMLST was 0.860 and 0.974, respectively. The median cgMLST distance between isolates with the same MLVA type was 27 alleles (interquartile range [IQR]:17 to 34 alleles), and 2 alleles within cgMLST clusters (IQR:1-5 alleles). This study shows the higher discriminatory power of WGS over MLVA and a poor concordance between both typing methods regarding clustering of *S.* Enteritidis isolates.

**IMPORTANCE** *Salmonella* is the most frequently reported agent causing foodborne outbreaks and the second most common zoonoses in the European Union. The incidence of the most dominant serotype Enteritidis has increased in recent years. To differentiate between *Salmonella* isolates, traditional typing methods such as pulsed-field gel electrophoresis (PFGE) and multiple locus variable-number tandem repeat analyses (MLVA) are increasingly replaced with whole-genome sequencing (WGS). This study compared MLVA and WGS-based core-genome Multilocus Sequence Typing (cgMLST) as typing tools for *S.* Enteritidis isolates that were collected as part of the national *Salmonella* surveillance in the Netherlands. We found a higher discriminatory power of WGS-based cgMLST over MLVA, as well as a poor concordance between both typing methods regarding clustering of *S.* Enteritidis isolates. This is especially relevant for cluster delineation in outbreak investigations and confirmation of the outbreak source in trace-back investigations.

**KEYWORDS** *Salmonella*, clustering, multiple locus variable-number tandem repeat analyses, whole-genome sequencing

Address correspondence to Roan Pijnacker, roan.pijnacker@rivm.nl.

The authors declare no conflict of interest.

*S*almonella is the second most common reported zoonosis in the European Union and the most important agent identified in foodborne outbreaks (1). In the Netherlands, the National Institute for Public Health and the Environment (RIVM) receives around 1,200 *Salmonella* isolates from human patients each year for further typing and national surveillance purposes through a countrywide network of medical microbiological laboratories. The dominant serotypes are Enteritidis and Typhimurium, including its monophasic variant, which are responsible for 28% and 24% of all *Salmonella enterica* subspecies *enterica* isolates, respectively (2). Despite a long-term decreasing trend, *S.* Enteritidis has grown in occurrence in recent years, not only in the Netherlands, but also Europe-wide (1).

Traditional serotyping based on the presence of specific combinations of O- and H-antigens resulting in distinct serovars (3, 4), has been the gold standard for *Salmonella* subtyping until recently. In recent years, however, many laboratories in public health and food safety agencies replaced traditional serotyping with *in-silico* serotyping using whole-genome sequencing (WGS), as a cost-effective, high-throughput, and rapid method (5, 6). High-resolution typing methods, such as WGS, are especially necessary for common serotypes Enteritidis and Typhimurium, to identify cases that are most likely linked to a common source of infection (i.e., "clusters"). Yet, to differentiate among *Salmonella* isolates, especially for outbreak detection of common serotypes, traditional typing methods still include pulsed-field gel electrophoresis (PFGE) and multiple locus variable-number tandem repeat analyses (MLVA). However, WGS has shown enhanced discriminatory power for *Salmonella* compared to PFGE or MLVA, with improved cluster delineation as well as microbiological confirmation of the outbreak source in trace-back investigations (7, 8). Moreover, WGS provides other relevant information about an isolate, such as the presence of resistance genes or virulence markers (5, 9).

Since January 2019, WGS has been implemented as the routine typing method of the RIVM for *S.* Enteritidis isolates. In parallel, molecular typing was performed based on MLVA. The objective of this study was to determine the concordance between MLVA and WGS as typing methods for *S.* Enteritidis isolates.

## RESULTS

***Salmonella* Enteritidis isolates and typing.** From January 2019 to March 2020, a total of 1,480 *Salmonella* isolates of human origin were received at the RIVM. Of these, 466 (31.5%) were serotyped as *S.* Enteritidis and confirmed as *S.* Enteritidis using *in-silico* determination with the Juno typing pipeline. A single isolate (0.2%) was excluded due to contamination. Of the remaining 465 isolates that were subjected to WGS, 26 (5.6%) sequences were disregarded because the WGS data contained more than 4% contaminant DNA. A total of 439 isolates that had both WGS and MLVA typing done were included for analyses. Most of the 439 isolates belonged to ST11 ($n = 420$, 96%), and the other to ST183 ($n = 9$), ST1925 ($n = 3$), ST3233 ($n = 2$), ST3406 ($n = 2$), ST366 ($n = 1$), ST4695 ($n = 1$), and one isolate had an unknown MLST type.

**Cluster duration and size.** For 439 isolates that had both MLVA and WGS typing done, 404 (92%) had the same MLVA type as at least one other case. These 404 isolates were resolved in 32 clusters (i.e., 32 MLVA types) with a median size of 4 isolates, but ranged from 2 to up to 141 isolates with the same MLVA type. Twelve of these 32 MLVA clusters had 2 isolates, 10 had 3 to 5 isolates, 2 had 5 to 9 isolates, and 8 had 10 isolates or more. Based on cgMLST, 313 isolates (71.1%) belonged to 48 clusters of at least two isolates based on cgMLST with a distance of ≤5 alleles (Table 1). The median cluster size was 3 isolates with a minimum of 2 and a maximum of 39 isolates within the same cluster. Twenty of these 48 cgMLST clusters had 2 isolates, 14 had 3 to 5 isolates, 5 had 6 to 9 isolates, and the remaining 9 clusters had 10 or more isolates. The ability to differentiate between all 439 isolates was calculated based on Simpson's diversity index, showing that cgMLST had a higher diversity of "types" than MLVA, with Simpson's diversity indexes of 0.974 and 0.860, respectively. The median cluster duration based on cgMLST was 2.6 months (range 0.0 to 14.1 months), which was lower than 5.1 months (range 0.1 to 13.8 months) based on MLVA. Larger clusters generally had a longer duration for both clustering methods.

The concordance between cgMLST and MLVA clusters is visualized in Fig. 1. Twenty out of the 48 cgMLST clusters (41.7%) consisted of 2 or more different MLVA types, with a

**TABLE 1** Cluster size and duration of *S. Enteritidis* isolates based on cgMLST and MLVA (*n* = 439), January 2019–March 2020

| | cgMLST | | | | | MLVA | | | | |
|---|---|---|---|---|---|---|---|---|---|---|
| | No. clusters | Isolates | | Duration in months | | No. clusters | Isolates | | Duration in months | |
| Size | N | N | % | Median | Max | N | N | % | Median | Max |
| Singleton | NA[a] | 126 | 28.7 | NA | NA | NA | 35 | 8.0 | NA | NA |
| 2 | 20 | 40 | 9.1 | 0.9 | 12.9 | 12 | 24 | 5.5 | 2.2 | 6.2 |
| 3−5 | 14 | 53 | 12.1 | 4.1 | 9.7 | 10 | 40 | 9.1 | 4.7 | 12.6 |
| 6−9 | 5 | 36 | 8.2 | 1.8 | 6.4 | 2 | 14 | 3.2 | 11.0 | 12.9 |
| 10−19 | 6 | 81 | 18.5 | 8.8 | 12.8 | 4 | 55 | 12.5 | 10.1 | 14.0 |
| 20−29 | 1 | 26 | 5.9 | 9.9 | 9.9 | 1 | 29 | 6.6 | 11.5 | 11.5 |
| 30−39 | 2 | 77 | 17.5 | 10.6 | 14.3 | 1 | 37 | 8.4 | 10.8 | 10.8 |
| 40−49 | 0 | 0 | 0 | NA | NA | 0 | 0 | 0 | NA | NA |
| 50−99 | 0 | 0 | 0 | NA | NA | 1 | 64 | 14.6 | 13.3 | 13.3 |
| 100+ | 0 | 0 | 0 | NA | NA | 1 | 141 | 32.1 | 13.8 | 13.8 |
| Total | 48 | 439 | 100 | 2.6 | 14.3 | 32 | 439 | 100 | 5.1 | 14.0 |

[a]NA, not applicable.

maximum of 6 different MLVA types within one cgMLST cluster. In turn, nine (28.1%) MLVA clusters consisted of 2 or more cgMLST clusters. The dominant MLVA types 02-10-07-03-02 (*n* = 141) and 02-11-07-03-02 (*n* = 64) consisted of 13 and 14 cgMLST clusters, respectively. Within MLVA type 02-10-07-03-02, there were several larger, cgMLST-defined clusters, e.g., 37, 21, 16, and 11 isolates. For MLVA type 02-11-07-03-02, however, there was only one larger cluster of 30 isolates and many other clusters no larger than 5 isolates. To determine the concordance between clusters identified based on MLVA and cgMLST, the FM index was calculated. The FM index was 0.34 on a scale from 0 to 1, where a higher value indicates a greater similarity between the two, which indicates a low concordance between the clustering based on MLVA and cgMLST.

**cgMLST distance versus MLVA distance.** The median and mean distance between isolates within cgMLST clusters was 2 alleles and 3 alleles, respectively (interquartile range [IQR]: 1 to 5 alleles), with a maximum allelic distance of 12 alleles (see Fig. 2). The overall median cgMLST distance between isolates with the same MLVA type was 27 alleles (IQR:17 to 34 alleles), with a maximum of 233 alleles. This varied per MLVA type, with the two largest MLVA clusters, i.e., MLVA type 02–10-07-03-02 (*n* = 141) and 02–11-07-03-02 (*n* = 64), having a median of 27 and 25 alleles distance between isolates, respectively (Fig. 3). The third and fourth largest MLVA clusters, namely, MLVA type 03–10-05-04-01 (*n* = 37) and 03–11-05-04-01 (*n* = 29), had a median of 60 and 57 alleles distance among isolates, respectively (see Fig. 3). The four most prevalent MLVA types had isolates with more than 100 alleles distance to all the other isolates with the same MLVA type. These were in total 10 isolates, of which two were singletons based on cgMLST (i.e., did not cluster with other isolates), and 8 clustered with at least one other isolate with a different MLVA type. Two isolates with distinct MLVA types, namely, 02-13-00-06-00 and 02-14-00-07-00, were separated by 51 alleles but had a minimum of 2,800 alleles difference from each of the other isolates in this study (see Fig. 4). Simpson's diversity indices for cgMLST within the four largest MLVA clusters were, in order of cluster size, 0.883, 0.770, 0.857, and 0.837, respectively. To determine whether some MLVA types had higher concordance with clusters based on cgMLST, the FM index was also calculated per MLVA type. Results showed that the four largest MLVA types and the other MLVA categorized as "Other" had similar concordance with cgMLST clustering, ranging between 0.341 and 0.479. Based on the Spearman rank correlation coefficient, the number of days between sampling dates of isolates with the same MLVA type was not associated with the distance in cgMLST alleles between them ($\rho$ = 0.02).

Isolates with 1 MLVA loci difference had a median difference in cgMLST alleles of 31 (IQR:25 to 52 alleles), those with 2 MLVA loci difference a median of 52 cgMLST alleles difference (IQR:34 to 82 alleles), 3 MLVA loci difference a median of 217 cgMLST alleles difference (IQR:90 to 237 alleles), 4 MLVA loci difference a median of 228 cgMLST alleles difference (IQR:220 to 237), and 5 MLVA loci difference a median of 227 cgMLST alleles difference

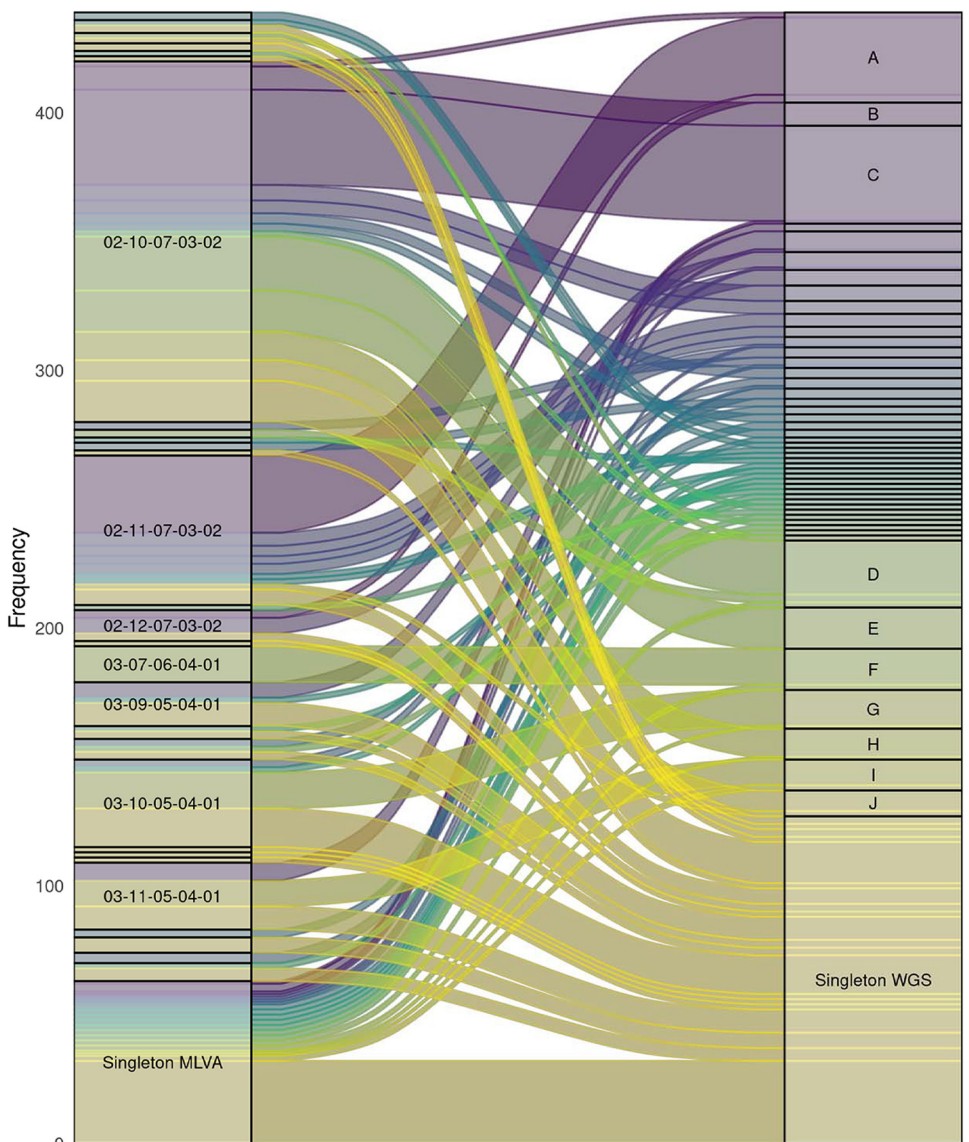

**FIG 1** Sankey diagram based on 439 *S.* Enteritidis isolates. The left side shows the isolates by MLVA type and the right side shows how they cluster based on cgMLST with a cutoff ≤5 alleles (3,002 loci), ignoring pairwise missing loci. "Singleton" MLVA types were those that were identified in one isolate only. The width of the blocks corresponds to the number of isolates. Only clusters with at least 10 isolates are labeled.

(IQR:222 to 235) (see Fig. 4). The Spearman's rank correlation coefficient between MLVA loci difference and difference in cgMLST alleles was 0.70 ($P < 0.001$).

**MLVA versus cgMLST in an outbreak investigation.** From June to July 24, 2019, a total of 21 cases with MLVA type 02-10-07-03-02 were reported, which was higher than what was usually observed during the same time period in 2016 to 2018, with 4, 8, and 10 cases, respectively. An outbreak investigation was initiated, where cases reported since June were interviewed retrospectively. Of the initial 21 cases, the majority (12/17, 71%) with WGS results available belonged to the outbreak based on cgMLST (Fig. 5). Therefore, but also to reduce recall bias, it was decided to interview future cases if they had the outbreak MLVA profile, and not wait for WGS results to become available 1 week later. However, in the following month, the majority of cases (9/12, 75%) would not belong to the outbreak based on cgMLST, after which was decided to guide patient interviews based on cgMLST. In total, 101 cases with the outbreak MLVA profile were reported between June and December 2019, of which 37 (37%) were outbreak-related based on cgMLST.

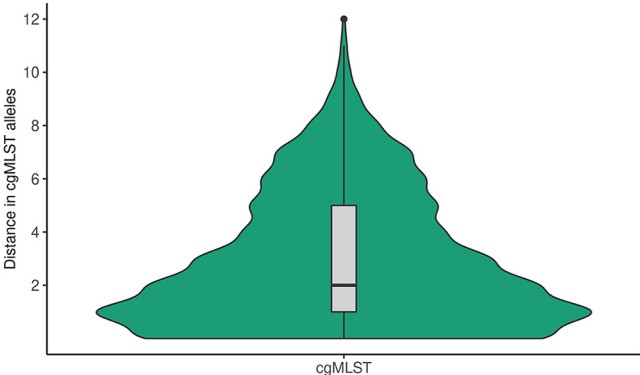

**FIG 2** Violin plot of the distance in cgMLST alleles between isolates within the same cgMLST cluster (*n* = 313). This includes 313 isolates that were linked to at least one other isolate based on a distance of ≤5 alleles.

One case with a different MLVA type, namely, 02-11-07-03-02, also belonged to the outbreak based on cgMLST. Although case interviews in the Netherlands did not identify a specific food item, the German authorities notified us about a household cluster of four *S.* Enteritidis cases that fell ill in November 2019 and were linked to the Dutch outbreak strain based on cgMLST. The German authorities reported that they presumable got ill though consumption of eggs that originated from a Dutch laying hen farm. Microbiological confirmation of eggs as the outbreak source was no longer possible since control measures were already undertaken by the implicated farm in December 2019. However, it was still deemed the most likely outbreak source because for the majority of cases in the Netherlands for which questionnaire data were available, 22/25 (88%) could be linked to supermarket chains that were supplied with eggs from the implicated laying hen farm. Of the 12 cases with the outbreak MLVA profile that were interviewed but were not part of the outbreak based on cgMLST, only three (25%) could be linked to the implicated supermarket chains. Moreover, no new cases were observed after control measures were taken by the implicated farm.

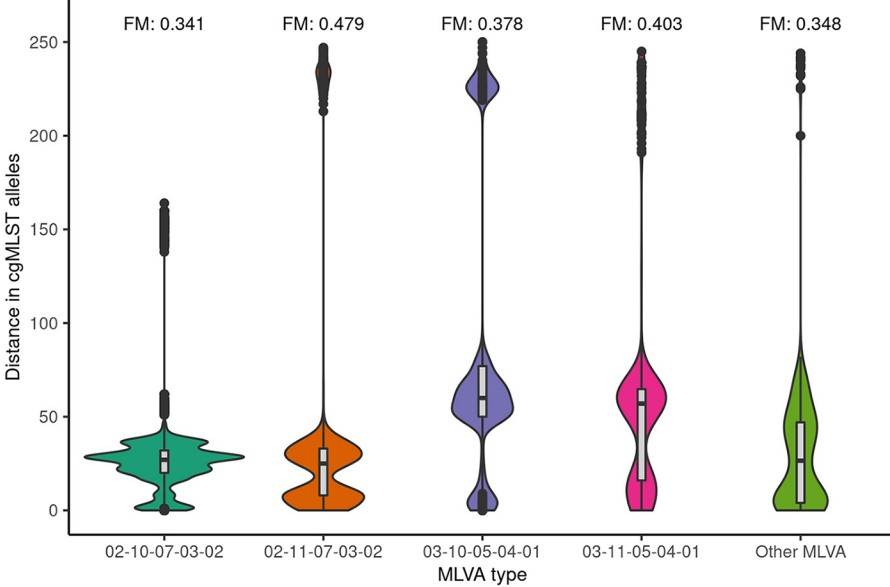

**FIG 3** Violin plot of the distance in cgMLST alleles between isolates by MLVA type. This included 404 isolates with MLVA types 02-10-07-03-02 (*n* = 141), 02-11-07-03-02 (*n* = 64), 03-10-05-04-01 (*n* = 37) and 03-11-05-04-01 (*n* = 29), and other MLVA types grouped as "Other MLVA." Isolates with a unique MLVA type (i.e., not found in any of the other isolates) were excluded. FM = Fowlkes-Mallows index (FM).

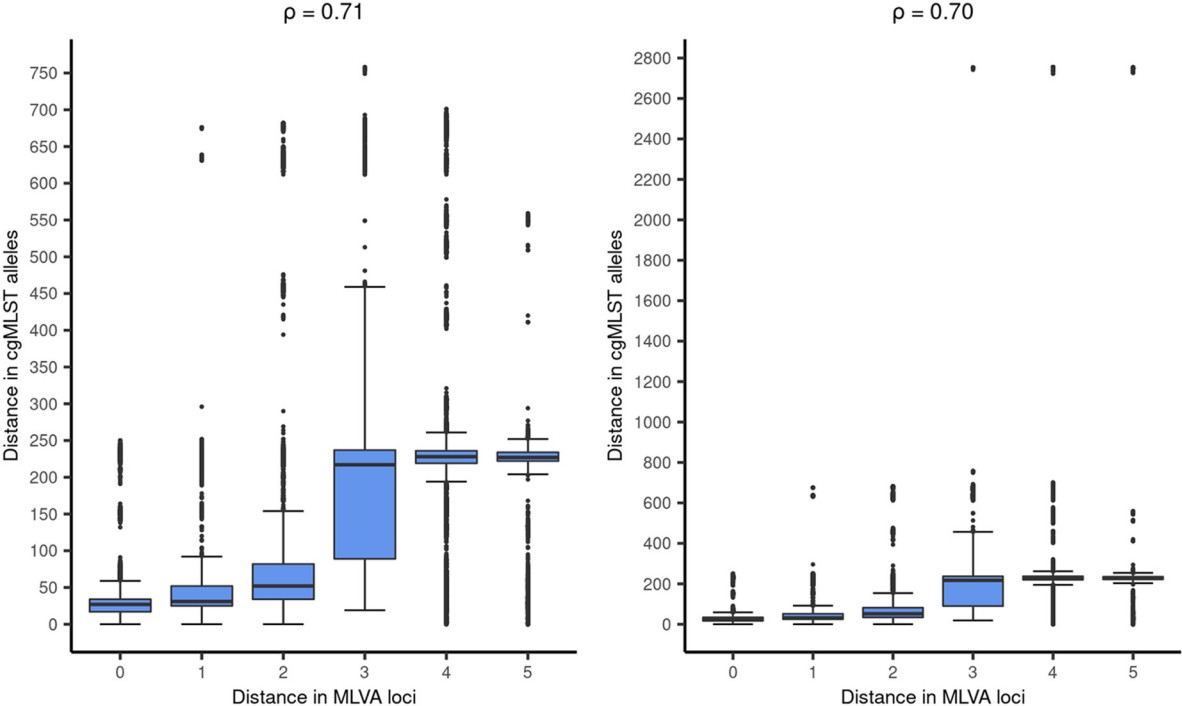

**FIG 4** Correlation between the number of MLVA loci difference and the difference in cgMLST alleles. The correlation between distance in MLVA loci and cgMLST alleles was calculated using the Spearman rank correlation coefficient ($\rho$). In the left panel, two isolates with over 2,800 allelic distance to the other isolates were excluded to improve the plot's readability. In the right panel, these isolates were included.

## DISCUSSION

WGS is rapidly replacing traditional typing methods, such as MLVA and PFGE, to differentiate between *Salmonella* isolates, especially in outbreak investigations. Here, we described the concordance between clusters identified by cgMLST and MLVA for serotype Enteritidis, based on 439 isolates received from January 2019 to March 2020 as part of the national laboratory system for *Salmonella* in the Netherlands. The overall concordance between the two typing methods was low, and a higher discriminatory power was observed for cgMLST in comparison to MLVA.

The FM-index between clustering based on MLVA compared with cgMLST was 0.34 on a scale of 0 to 1, where a higher value indicates greater similarity between clustering. cgMLST

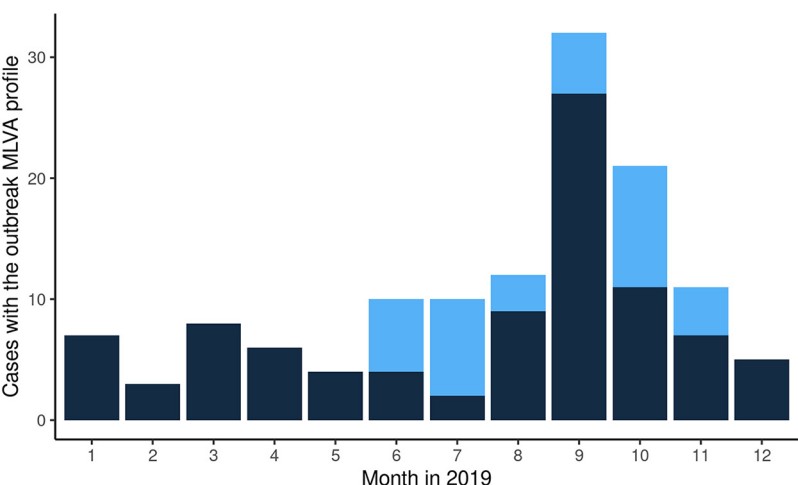

**FIG 5** Number of cases with the outbreak MLVA profile (02-10-07-03-02) that were part of the outbreak cluster based on cgMLST by month, 2019. Light blue = part of the outbreak based on cgMLST. Dark blue = not part of the outbreak based on cgMLST.

identified a larger number of clusters with fewer number of isolates compared with MLVA. We hypothesized that the discriminatory power of the MLVA subtyping procedure might be lower for dominant MLVA types 02-10-07-03-02 and 02-11-07-03-02, who together make up almost half of the isolates, and higher for less prevalent MLVA types. However, the FM-index between MLVA types was almost identical compared with clustering based on cgMLST, indicating that the discriminatory power is similar for less prevalent and dominant MLVA types. The higher discriminatory power of WGS compared with traditional methods is consistent with studies conducted in Australia, Denmark, Tunisia, United Kingdom, and United States, which all identified benefits of WGS over PFGE or MLVA (8, 10–15). This has also been increasingly recognized in multicountry *Salmonella* outbreak investigations that employed both WGS and MLVA-based case definitions (16). The lower discriminatory power of MLVA likely also explains the longer cluster duration of MLVA clusters compared with cgMLST, as some isolates are wrongly assigned to a cluster. We also assessed whether MLVA typing would perform better when the time interval between sampling of isolates was short, which would be relevant for outbreaks that occur in a short time period. However, the concordance of cgMLST and MLVA with regard to cluster identification did not change with time interval between sampling of isolates.

Many countries still perform MLVA due to the higher material costs for WGS. However, one could argue that personnel costs is spent less efficiently when using MLVA due to inaccuracy in outbreak delineation compared to WGS. This means that patients that do not actually belong to the outbreak are included in the outbreak investigation and are interviewed to identify a common exposure among the isolates, thereby hampering the source tracing effort due to distortion of exposure data. Furthermore, WGS has applications beyond cluster delineation, such as identification of genes encoding antimicrobial resistance, virulence genes and serotyping, which further dilutes the per-sample costs of WGS. At the time of writing the manuscript, the costs for serotyping and MLVA combined and WGS currently stand at circa €106.50 and €143.00 per sample at the RIVM, respectively, including value-added tax (VAT) and laboratory personnel costs. These do not include costs for the maintenance of the necessary infrastructures, which is considerable for both methods. However, WGS automatic bioinformatic pipelines can also provide other relevant information as part of their output, such as the presence of virulence or resistance genes. WGS might therefore be more cost-effective than MLVA. The turnaround time for MLVA and WGS is also comparable, with approximately 5 working days from the day of sample receival.

In recent years, WGS has replaced MLVA as microbiological criterion in confirmed case definitions to determine which cases belong to the outbreak and which cases do not. In international outbreak investigation, WGS has proven to be superior to MLVA. Although MLVA is still used to define probable cases (17), it is likely that case definitions in *Salmonella* Enteritidis outbreaks will be solely based on WGS in the future, at least in countries where WGS facilities are available. For example, a Europe-wide outbreak of *S.* Enteritidis in 2021 that had been ongoing for several years was delineated using WGS only, without a MLVA-based case definition (18). Although this greatly increases outbreak delineation, it also poses challenges. For example, countries that do not perform WGS will not know whether they have cases related to the outbreak, especially because backwards compatibility of WGS with PFGE and MLVA is very limited (19). In contrast to MLVA, there is not one single pipeline used by different organizations, although a previous study showed that different single nucleotide polymorphisms (SNP) and allele-based typing (20) workflows used by several European public-health authorities identified concordant clustering of *S.* Enteritidis isolates (20). However, there is no consensus regarding nomenclature, hampering international communication on WGS results, but also at national level with policy makers and local health authorities. There have been attempts to facilitate communication of WGS results, such as the Public Health England (PHE) SNP pipeline, from which a SNP address can be derived (21). Another example is PulseNet International, which envisions a global network of public health laboratories that use WGS and employ the same nomenclature based on wgMLST (19). Despite these efforts, the issue of communicating WGS results has proven challenging and has not been

solved to date. Moreover, there are privacy considerations when sharing raw sequences across institutes and even sectors in order to prevent risks of privacy breaches (22).

There were two isolates with a distance of over 2,800 alleles with the other *S*. Enteritidis isolates. Given that the reference template to which it was compared had 3,002 targets, this finding was surprising. The isolates both had ST3406, which has to our knowledge, not been reported before. These isolates did not cluster with each other based on cgMLST and had a distance of 51 alleles between them. *In-silico* serotyping confirmed that these isolates belonged to subspecies *enterica* and revealed that they were not in possession of the *sdf* gene and could therefore belong to both serovar Enteritidis and Gallinarum as they share the same formula based on gene detection. However, traditional serotyping based on the presence of O- and H-antigens confirmed that they were serotype Enteritidis. Additionally, hierarchical clustering with the isolates in the *Salmonella* database of Enterobase to assess the global context, indicated that the two isolates belong to cluster 43000 on level H50, consisting of only *S*. Enteritidis isolates.

This study has several limitations. *Salmonella* surveillance in the Netherlands is sentinel, meaning that it collects isolates of a selection of laboratories with an estimated population coverage of 62% (23). As a result, cluster size and duration of MLVA and cgMLST are likely underestimated. Cluster duration is also likely to be longer because we only included isolates during a 14-month period where WGS and MLVA were performed in parallel. However, these limitations are unlikely to have influenced the concordance of clustering based on MLVA and cgMLST, which was the main aim of this study. The strength of this study is that it compared MLVA and cgMLST as typing methods on all isolates that were serotyped as Enteritidis, regardless of whether they were outbreak-related. In contrast, most studies comparing WGS with PFGE or MLVA did so retrospectively or prospectively in outbreak settings only.

In conclusion, this study confirms the higher discriminatory power of WGS-based cgMLST over MLVA as well as a poor concordance between both typing methods regarding clustering. Although WGS has greatly improved our ability to delineate clusters, there are still hurdles to overcome. This is especially the case for international communication of clusters because there is no consensus on nomenclature used by different institutes, as well as the limited backwards comparability with traditional typing methods such as PFGE and MLVA. However, these shortcomings are outweighed by the benefits of WGS.

## MATERIALS AND METHODS

**Study population.** This study used pure cultures of *Salmonella* isolates originating from humans that were sent by medical microbiological laboratories to the RIVM from January 2019 to March 2020 for further typing and characterization as part of the national laboratory surveillance system for *Salmonella*. Molecular screening of their serotypes was performed using the Luminex xMAP *Salmonella* Serotyping assay kit according to the manufacturer's instructions. After molecular screening, confirmative classical slide agglutination with *Salmonella* O- and H-antisera according to the White-Kauffmann-Le Minor scheme was applied (3, 4). Here, we only included isolates that were serotyped as *S*. Enteritidis and subtyped using MLVA and WGS. If multiple *S*. Enteritidis isolates were isolated from the same patient, only the first isolate was included regardless of the origin of that isolate.

**MLVA.** For MLVA, five previously identified loci were amplified as described (24). Capillary electrophoresis for fragment analysis was outsourced to a commercial company (BaseClear B.V., Leiden, the Netherlands) and resulting profiles were analyzed in-house in Bionumerics (version 7.6.3.). MLVA profiles were depicted in allele strings SENTR7_SENTR5_SENTR6_SENTR4_SE-3. Isolates with the same MLVA profile were considered a cluster. Distance between isolates was determined as the number of loci at which they differ, also referred to as locus variants, ranging between 0 and 5.

**Whole-genome sequencing.** For WGS analysis, DNA was extracted using the GenElute Bacterial Genomic DNA kit (Sigma-Aldrich, Inc.) according to the manufacturer's protocol. Library preparation was executed using the Illumina Nextera XT DNA Library Prep kit or the Illumina DNA Prep (Illumina, Inc.). Sequencing was performed on a Illumina NextSeq machine, resulting in $2 \times 150$ bp reads. Sequence data were submitted to the European Nucleotide Archive (ENA), study number PRJEB54672. Reads were processed using the in-house developed pipeline "Juno assembly," consisting of read trimming, extensive QC control of raw reads, trimmed reads and assemblies, and *de novo* assembly (25). Only raw reads with a phred quality score >30 and resulting novo assemblies with a total length between 4,540,000-5,210,000 bp, $N_{50} > 10,000$ bp, GC percentage between 51.6 and 52.3%, number of contigs <300, average coverage >10, genome completeness >96%, and a contamination <4% passed our quality criteria. Using the output of Juno assembly, an in-house developed pipeline "Juno typing" based on SeqSero2 (26) was applied for *in-silico Salmonella* serotyping (27). *De novo* assemblies were imported into Ridom SeqSphere, where allelic profiles were determined using the Enterobase *Salmonella enterica* v2.0 core genome Multilocus Sequence Typing (cgMLST) scheme comprising 3,002 loci, as well as with

the 7-locus MLST scheme. On the same platform, distance matrices were calculated from the allelic profiles using a Hamming distance, ignoring pairwise missing loci. Clusters were defined from these matrices using a hierarchical agglomerative clustering approach with single linkage. For cluster definition, a distance of ≤5 alleles was used as cutoff.

**Analysis.** Clustering of isolates based on MLVA and cgMLST was compared with regard to cluster size and cluster duration in months. The Fowlkes-Mallows (FM) index was calculated to determine the concordance between the two clustering methods (28). To identify whether some MLVA types had higher concordance with clustering based on cgMLST, the FM index was also calculated for the four most common MLVA types and the other MLVA types grouped as "Other." The discriminatory power of the two typing methods was assessed using Simpson's diversity index to measure their differences in diversity (29). For each of the clustering methods, the median cgMLST Hamming distance within their clusters was determined, as well as the interquartile range (IQR). The Spearman rank correlation coefficient was used to determine the correlation between MLVA and cgMLST loci distances among isolates, as well as whether the number of days between two isolates with the same MLVA type was correlated with the number of cgMLST alleles between them. We also compare the use of MLVA and cgMLST typing in an outbreak investigation that was performed during the study period. Data analysis was performed in R. The FM index was calculated using the "pci" function of the profdpm package v.3.

## ACKNOWLEDGMENTS

We thank Gerhard Falkenhorst and Sandra Simon from the Robert Koch Institute (RKI), Germany, for notifying us about their *S*. Enteritidis family cluster that matched with the outbreak strain in the Netherlands. Without their help, we would likely not have known the outbreak vehicle of infection.

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
