## [Reviewer comments · Microbiology Spectrum]

Microbiology Spectrum

Comparing multiple locus variable-number tandem repeat analyses with whole genome sequencing as typing method for *Salmonella* Enteritidis surveillance in the Netherlands, January 2019 to March 2020

Roan Pijnacker, Maaïke van den Beld, Kim van der Zwaluw, Anjo Verbruggen, Claudia Coipan, Alejandra Segura, Lapo Mughini-Gras, Eelco Franz, and Thijs Bosch

Corresponding Author(s): Roan Pijnacker, National Institute for Public Health and the Environment (RIVM), Centre for Infectious Disease Control

Review Timeline:

Submission Date:	April 19, 2022
Editorial Decision:	June 17, 2022
Revision Received:	August 3, 2022
Accepted:	August 29, 2022

Editor: Sadjia Bekal

Reviewer(s): Disclosure of reviewer identity is with reference to reviewer comments included in decision letter(s). The following individuals involved in review of your submission have agreed to reveal their identity: Genevieve Labbe (Reviewer #2); Dele Ogunremi (Reviewer #3)

Transaction Report:

DOI: <https://doi.org/10.1128/spectrum.01375-22>

June 17, 2022

Dr. Roan Pijnacker
National Institute for Public Health and the Environment (RIVM), Centre for Infectious Disease Control
Bilthoven
Netherlands

Re: Spectrum01375-22 (Comparing multiple locus variable-number tandem repeat analyses with whole genome sequencing as typing method for Salmonella Enteritidis surveillance in the Netherlands, January 2019 to March 2020)

Dear Dr. Roan Pijnacker:

Link Not Available

Sincerely,

Sadjia Bekal

Journals Department
Reviewer comments:

Reviewer #1 (Comments for the Author):

This study is a simple measure of concordance and discriminatory power of WGS, which has been studied perviously. Nevertheless, having this comparison performed and published is indeed important for public health laboratories to be able to justify changes to accredited tests and operational procedures. The paper is clear and concise.

Major Comments

Lack of epidemiological context: The paper is lacking any epidemiological context, which is the basis for laboratories to change from molecular subtyping (PFGE, MLVA) to WGS. The authors rightly comment that their study benefits from not being restricted to an outbreak perspective only; however, an important concordance to measure is that of the laboratory test and the public

health outcome it is attempting to mitigate. This paper would be significantly strengthened by the addition of some epidemiological context. Surely within these approximately 400 isolates there were some potential outbreaks investigated, did the cases ruled in or ruled out change when cgMLST was used instead of MLVA? With higher discriminatory power and very different clustering of WGS compared to MLVA, what improvement did this bring to actual public health laboratory function during the period of parallel study?

Definition of cgMLST cluster: Why was a cut-off of 5 alleles or less used for clustering via cgMLST? Given that all of the results in this paper: concordance, clustering, discriminatory power will change depending on the allele threshold, this is a critical issue. (i.e., the findings would be different if the threshold was set at 10 alleles or less, or 2 alleles or less). Justification for this threshold must be provided s measured based on the categorization of isolates into clusters.

Data: the sequence data should be made available upon publication.

Minor Comments

Figure 1: it is not clear what "singleton MLVA" refers to, is that entire block comprised of different MLVA profiles? If so, why are there many isolates in the same shade of brown? Does the width of the colour blocks/lines in the Sankey diagram represent number of isolates? Please add more description to the legend.

Line 288: what measures were taken to ensure that the sequenced isolates that were very distant were not mixed cultures/contaminated? This anomaly should be more thoroughly investigated. For example, were they resequenced? Were additional pipelines run to check for quality problems? Were there any other signals that indicated something wasn't quite right with these isolates? What is the provenance of these two isolates?

Reviewer #2 (Comments for the Author):

This is a very well crafted manuscript, with sound results. The WGS data from the 439 Salmonella Enteritidis isolates used in this study should ideally be deposited in a public nucleotide data archive (e.g. if deposited in NCBI, the NCBI SRA BioProject number should be provided in the manuscript)

Reviewer #3 (Comments for the Author):

See attached review comments

Staff Comments:

Preparing Revision Guidelines

Please return the manuscript within 60 days; if you cannot complete the modification within this time period, please contact me. If you do not wish to modify the manuscript and prefer to submit it to another journal, please notify me of your decision immediately so that the manuscript may be formally withdrawn from consideration by Microbiology Spectrum.

If your manuscript is accepted for publication, you will be contacted separately about payment when the proofs are issued; please follow the instructions in that e-mail. Arrangements for payment must be made before your article is published. For a

complete list of **Publication Fees**, including supplemental material costs, please visit our website.

Review of the manuscript "Comparing multiple locus variable-number tandem repeat analyses with whole genome sequencing as typing method for *Salmonella* Enteritidis surveillance in the Netherlands, January 2019 to March 2020" (control no. Spectrum01375-22).

Summary of the major findings of the article:

This study demonstrates that WGS analysis provides higher resolution than MLVA typing for public health surveillance and outbreak detection of *Salmonella* Enteritidis, based on parallel typing of 439 isolates collected from human patients in the Netherlands over a 13 months period. The authors also show that there is a poor correlation between the clustering results obtained by the two subtyping methods.

Reviewer's overall impression of the paper:

The manuscript is well written, the methods are sound, the findings are presented in a clear and concise manner, and the conclusions are also supported by other studies done previously for different *Salmonella* serovars. This reviewer is also aware that similar results were obtained in a (still unpublished) study of Canadian isolates of *Salmonella* Enteritidis at the National Microbiology Laboratory in Canada (*personal communication*).

The findings presented in this study are very relevant for public health organizations, as the authors pointed out, for MLVA inaccuracy in outbreak delineation (compared to WGS) can hamper the source tracing efforts.

Major shortcoming:

The WGS data from the 439 *Salmonella* Enteritidis isolates used in this study should ideally be deposited in a public nucleotide data archive such as ENA or NCBI

Comments:

- #1 -The WGS data should be deposited in a public nucleotide archive, and the reference ID(s) should be provided in the manuscript
- #2 - Lines 97-99 "*Fragment analysis was performed externally*": can this be explained further? Were the methods used identical to those described in Reference #10 (Hopkins et al.)?
- #3 -Minor corrections/clarifications should be made to the text:
 - Author affiliations – a to g; and i and j, are all the same affiliation and should all be collapsed into one affiliation.
 - Line 52: typo, italicize *S.* in "*S.* Enteritidis"

- Lines 141-142: the sentence *"The remaining 465 isolates were subjected to WGS, after which 439 were left because 26 (5.6%) sequences were disregarded because the WGS data contained more than 4% contaminant DNA."* should be rephrased in order to avoid using the word "because" twice in the same sentence.

-Line 260: "VAT" should be spelled out

-Line 269: typo, replace "te" by "to": *"WGS has proven te be superior"*

-Line 276: typo, extra "s" in the word pipeline: *"one single pipelines"*

The authors would like to thank the reviewers for their thoughtful comments. Please find below our answers to the reviewer's specific comments.

Reviewer #1

This study is a simple measure of concordance and discriminatory power of WGS, which has been studied perviously. Nevertheless, having this comparison performed and published is indeed important for public health laboratories to be able to justify changes to accredited tests and operational procedures. The paper is clear and concise.

Major Comments

1. Lack of epidemiological context: The paper is lacking any epidemiological context, which is the basis for laboratories to change from molecular subtyping (PFGE, MLVA) to WGS. The authors rightly comment that their study benefits from not being restricted to an outbreak perspective only; however, an important concordance to measure is that of the laboratory test and the public health outcome it is attempting to mitigate. This paper would be significantly strengthened by the addition of some epidemiological context. Surely within these approximately 400 isolates there were some potential outbreaks investigated, did the cases ruled in or ruled out change when cgMLST was used instead of MLVA? With higher discriminatory power and very different clustering of WGS compared to MLVA, what improvement did this bring to actual public health laboratory function during the period of parallel study?

Answer: We agree with the reviewer that epidemiological context would strengthen the manuscript. In the Netherlands, we only collect epidemiological data (i.e. exposure data on food consumption) during outbreak investigations. Therefore, we added a new paragraph to the manuscript that describes the use of both MLVA and cgMLST typing in an outbreak investigation that was performed during the study period. It describes how MLVA typing was useful to determine which cases belonged to the outbreak during the first few weeks of the investigation, but also that an increasing number of cases with the outbreak MVLVA profile were found not to be part of the outbreak based on cgMLST as the outbreak progressed. Hence, case interview were no longer guided by the outbreak MLVA profile, but cgMLST cluster instead. Furthermore, case interviews also indicated that exposure data would be "diluted" if cases would be interviewed based on MLVA typing only, as they often did not buy their groceries as the implicated supermarket chains.

2. Definition of cgMLST cluster: Why was a cut-off of 5 alleles or less used for clustering via cgMLST? Given that all of the results in this paper: concordance, clustering, discriminatory power will change depending on the allele threshold, this is a critical issue. (i.e., the findings would be different if the threshold was set at 10 alleles or less, or 2 alleles or less). Justification for this threshold must be provided s measured based on the categorization of isolates into clusters.

Answer: In European multi-country outbreak investigations where clusters were defined based on cgMLST, the threshold was always set at 5 alleles ([Multi-country outbreak of monophasic Salmonella Typhimurium sequence type \(ST\) 34 linked to chocolate products \(europa.eu\)](https://www.eurosurveillance.org/ViewArticle.aspx?pubId=34)) and [ROA Salmonella Enteritidis ST11 infections linked to eggs \(europa.eu\)](https://www.eurosurveillance.org/ViewArticle.aspx?pubId=34)), similar to what we used in our study. Moreover, the Food- and Waterborne Disease Antimicrobial Resistance – Reference Laboratory Capacity has recently recommended the same cut-off of 5 alleles for [Salmonella 9 fwd-amr-reflabcap-wgs-protocol-version1.pdf \(fwdamr-reflabcap.eu\)](https://www.fwd-amr-reflabcap-wgs-protocol-version1.pdf).

3. Data: the sequence data should be made available upon publication.

Answer: We agree with the reviewer and made sequence data available here: European Nucleotide Archive (ENA), study number PRJEB54672. This is now also mentioned in the manuscript.

Minor Comments

4. Figure 1: it is not clear what "singleton MLVA" refers to, is that entire block comprised of different MLVA profiles? If so, why are there many isolates in the same shade of brown? Does the width of the colour blocks/lines in the Sankey diagram represent number of isolates? Please add more description to the legend.

Answer: We now clarified in the description of the Sankey diagram that singleton MLVA refers to MLVA types that were identified in one isolate only. The isolates with the same shade of brown are MLVA types that were identified in only one isolate and were also singletons based on cgMLST, meaning that they were not part of a cluster based on cgMLST. The width of the blocks indeed correspond to the number of isolates, where the wider the block, the more isolates it represents.

5. Line 288: what measures were taken to ensure that the sequenced isolates that were very distant were not mixed cultures/contaminated? This anomaly should be more thoroughly investigated. For example, were they resequenced? Were additional pipelines run to check for quality problems? Were there any other signals that indicated something wasn't quite right with these isolates? What is the provenance of these two isolates?

Answer: There were multiple measures taken to ensure that the distant isolates were not mixed or contaminated cultures. First, it was ensured that the sequence of these two isolates were of good quality as with all the other isolates as described in the methods. In our in-house developed pipeline, an automatic extensive quality control is performed using MultiQC, QUAST and Bbtools to assess the quality of our reads and *de novo* assemblies. Passing criteria are mentioned in the method section, paragraph "whole-genome sequencing". Contamination control was automatically performed using CheckM and Kraken2. Additionally, SeqSero2 in micro-assembly mode was used which allows for detection of serotype mixtures of *Salmonella* sequences. None of these tools indicated bad quality, intermixing serotypes or contamination. Both isolates are of human origin and isolated from stool samples of different patients. They both lack the *sdf*-gene, and were therefore confirmed as serotype Enteritidis (9:g,m:-) with phenotypic typing using classical agglutination techniques as described in the discussion. Because of the comments of the reviewers, the distant isolates were imported into Enterobase to assess their global context. Hierarchical clustering of both isolates in the database of Enterobase indicated that both isolates belong to hierarchical cluster 43000 on level H50, consisting of all *Salmonella* Enteritidis isolates. Both isolates are clustering in different clusters on levels H20, H10 and H5, but cluster only with isolates determined as *Salmonella* Enteritidis by different institutes in Europe and United States. The results of the hierarchical clustering are added to the discussion section.

Reviewer #2

Summary of the major findings of the article:

This study demonstrates that WGS analysis provides higher resolution than MLVA typing for public health surveillance and outbreak detection of *Salmonella* Enteritidis, based on parallel typing of 439 isolates collected from human patients in the Netherlands over a 13 months period. The authors also show that there is a poor correlation between the clustering results obtained by the two subtyping methods.

Reviewer's overall impression of the paper:

The manuscript is well written, the methods are sound, the findings are presented in a clear and concise manner, and the conclusions are also supported by other studies done previously for different Salmonella serovars. This reviewer is also aware that similar results were obtained in a (still unpublished) study of Canadian isolates of Salmonella Enteritidis at the National Microbiology Laboratory in Canada (personal communication).

The findings presented in this study are very relevant for public health organizations, as the authors pointed out, for MLVA inaccuracy in outbreak delineation (compared to WGS) can hamper the source tracing efforts.

Major shortcoming:

The WGS data from the 439 Salmonella Enteritidis isolates used in this study should ideally be deposited in a public nucleotide data archive such as ENA or NCBI

Comments:

6. The WGS data should be deposited in a public nucleotide archive, and the reference ID(s) should be provided in the manuscript

Answer: We agree with the reviewer and made sequence data available here: European Nucleotide Archive (ENA), study number PRJEB54672. This is now also mentioned in the manuscript.

7. Lines 97-99 "Fragment analysis was performed externally": can this be explained further? Were the methods used identical to those described in Reference #10 (Hopkins et al.)?

Answer: Only capillary electrophoresis to perform fragment length analysis was outsourced to a commercial company, because equipment is not available at our institute. Amplification, preparation for fragment analysis and analysis of fragments were performed at our institute identical as described in Hopkins et al. This is clarified in methods section.

Minor corrections/clarifications should be made to the text:

8. Author affiliations – a to g; and i and j, are all the same affiliation and should all be collapsed into one affiliation.

Answer: The article guidelines stated that the affiliations of each author should be stated separately.

9. Line 52: typo, italicize S. in "S. Enteritidis"

Answer: Thank you. Amended as suggested.

10. Lines 141-142: the sentence "The remaining 465 isolates were subjected to WGS, after which 439 were left because 26 (5.6%) sequences were disregarded because the WGS data contained more than 4% contaminant DNA." should be rephrased in order to avoid using the word "because" twice in the same sentence.

Answer: In addition to the reviewers suggestions, we rephrased the sentence before, which also contained the word "because", as well as the following sentence to improve readability. It now reads as: " A single isolate (0.2%) was excluded due to contamination. Of the remaining 465 isolates that were subjected to WGS, 26 (5.6%) sequences were disregarded because the WGS data contained more than 4% contaminant DNA. A total of 439 isolates that had both WGS and MLVA typing done were included for analyses."

11. Line 260: "VAT" should be spelled out
 12. Line 269: typo, replace "te" by "to": "WGS has proven te be superior"
 13. Line 276: typo, extra "s" in the word pipeline: "one single pipelines"
- Answer:* Comment #11-12 amended as suggested.

Reviewer #3

General comments

Authors have addressed an important topic on subspecies characterization of the prevalent foodborne pathogen, *Salmonella* Enteritidis, and have shed light on the superior discriminatory ability of the whole genome sequencing approach over the more traditional Multiple Locus Variable-number tandem repeat analyses. The paper is well-written and contains useful and pertinent information for experts in the field.

There are two areas that could be further improved upon, namely the description of the methods and observations of two atypical isolates as outlined below.

14. A number of sentences outlined under specific comments could be enhanced to allow the reader to appreciate the approaches taken and for an experimenter to readily reproduce the results (See lines 93-132, under Specific comments). Central to this concern is the public availability of the data which is not mentioned in the paper. Authors should also described the process of determining the number of allelic differences by the WGS approach (e.g., compiling data from Enterobase).

Answer: We have addressed the specific comments of the reviewer below. The sequence data have been made available here: European Nucleotide Archive (ENA), study number PRJEB54672. We believe that the process of determining the allelic distance between isolates was already described in the manuscript. We here copy an exempt of the methods sections: "*De novo* assemblies were imported into Ridom SeqSphere, where allelic profiles were determined using the Enterobase *Salmonella enterica* v2.0 core genome Multilocus Sequence Typing (cgMLST) scheme comprising 3,002 loci, as well as with the 7-locus MLST scheme. On the same platform, distance matrices were calculated from the allelic profiles using a Hamming distance, ignoring pairwise missing loci. Clusters were defined from these matrices using a hierarchical agglomerative clustering approach with single linkage."

15. The designation of the two atypical isolates as Enteritidis is not convincing (Line 291-295). The lack of the *sdf* gene, being the single most distinguishing feature of Enteritidis (Agron et al., Appl Environ Microbiol 67:4984, 2001) argues against this serovar classification, as also does the large number of allelic differences when compared to the other Enteritidis isolates in the study. SeqSero2 used in this study is reliable but the accuracy, as with many other tools, is less than perfect. The discussion should present the contrasting evidence. Authors should investigate and develop further evidence for subspecies characterization of the organisms, e.g., a phylogenetic tree for as many *Salmonella* serovars as possible, each with multiple representatives to see the closest serovar relative to the atypical isolates, or by prophage analysis (Mottawea et al., Front. Microbiol. 9:836, 2018).

Answer: We have investigated these two atypical isolates extensively. Good quality of sequences and absence of serotype intermixing or contamination was ensured by using our in-house developed pipeline, consisting of MultiQC, QUAST and Bbtools to assess the quality of our reads and *de novo* assemblies. Passing criteria are mentioned in the method section, paragraph "whole-genome sequencing". Contamination control was automatically performed using CheckM and Kraken2. Additionally, SeqSero2 in micro-assembly mode was used which allows for detection of serotype mixtures of *Salmonella* sequences. We have validated the SeqSero2 tool according to ISO 15189 standards and assessed an accuracy of 99.4% compared to phenotypical

serotyping using 509 Salmonella isolates comprising 181 serotypes. None of the deviant isolates were Salmonella Enteritidis isolates. Because the sdf gene was lacking, these two isolates were confirmed with traditional agglutination techniques as described in discussion, resulting in O9, presence of 1st H-phase g,m and absence of 2nd H-phase, indicating towards the serovar Enteritidis. Because of the comments of the reviewers, the distant isolates were imported into Enterobase to assess their global context. Hierarchical clustering of both isolates in the database of Enterobase indicated that both isolates belong to hierarchical cluster 43000 on level H50, consisting of all Salmonella Enteritidis isolates. Both isolates are clustering in different clusters on levels H20, H10 and H5, but cluster only with isolates determined as Salmonella Enteritidis by different institutes in Europe and United States. The results of the hierarchical clustering are added to the discussion section.

Specific comments

16. Line 37: resolved in? 92% typed as 32 clusters

17. Line 38: resolved in?.....isolates typed as 48 clusters.....

18. Line 48: European Union.

Answer: Comment #16-18 are amended as suggested.

19. Line 59: important? and the most prevalent bacterial

Answer: Amended as “*Salmonella* is the most frequently reported agent causing...”

20. Line 93/94: Clarify whether “multiple isolates” are from the same sample or different samples from the same patient.

Answer: Regardless of the origin of the isolates, the first isolate was included in case multiple isolates were received from one patient. This is now clarified in methods.

21. Line 97: five previously identified loci were amplified as described (10).

22. Line 107-109: Sentence indicate that genome assemblies are repeated. The order should be: trim, QC and assembly. Rephrase.

Answer: Comment #21 and #22 amended as suggested

23. Line 132: sampling of isolates, and cluster duration as the time between the sampling of the first and last isolates in the same cluster.

Answer: Apologies, it was not written clearly. We meant the correlation between the number of days between sampling dates of isolates with the same MLVA profile and the distance in cgMLST alleles between them. We rationale behind this analysis is that we thought that two isolates with the same MLVA type would maybe be more likely te belong to the same cluster based on cgMLST when they had a shorter time interval between them. It is now described as “The Spearman rank correlation coefficient was used to determine the correlation between [...], as well as whether the number of days between two isolates with the same MLVA type was correlated with the number of cgMLST alleles between them.”

24. Line 170:several larger, cgMLST-defined clusters, e.g., 37,.....

25. Line 171-172: one larger cgMLST-defined cluster of 30 isolates and many other clusters no larger.....

26. Line 189:largest MLVA cluster namely 03-10.....

27. Line 190:..... alleles distance among isolates.....

Answer: Comment #24-27 are amended as suggested.

28. Line 192-194: The sentence is difficult to follow. Rephrase for clarity. It is not clear if these are singletons by one of the two methods.

Answer: Rephrased as “These were in total 10 isolates, of which two were singletons based on cgMLST (i.e. did not cluster with other isolates), and 8 clustered with at least one other isolate with a different MLVA type.”

29. Line 194:Two isolates with distinct MLVA types, namely 02-13-00-06-00 type and MLVA 02—14- 00-07-00 were separated by 51 alleles but had a minimum of 2,800 alleles difference from each of the other isolates in this study (Fig 4).

30. Line 196: within the four largest.....

Answer: Comment #29 and #30 amended as suggested

31. Line 229 vs 267 should agree: replacing vs. has replaced.

Answer: MLVA is still being used as typing tool for *Salmonella* in many countries, that’s why we prefer “is replacing” in line 229. However, in outbreak investigations specifically (line 267), it has already replaced MLVA typing in developed countries. Therefore, we prefer to keep it as it is.

32. Line 238: power of the MLVA subtyping procedure

Answer: Amended as suggested.

33. Line 240-241: How this observation contributes to addressing the hypothesis in previous sentence should be described. Does it clarify or nullify the hypothesis?

Answer: These findings indicate that the discriminatory power of MLVA typing is similar for less prevalent MLVA types and dominant MLVA types. Now written as: “[...] clustering based on cgMLST, indicating that the discriminatory power is similar for less prevalent and dominant MLVA types.”

34. Line 249: Are the data supporting this conclusion shown anywhere in the manuscript? If not, state “data not shown”.

Answer: Yes, these data were described in line 132, but we understand that it was not clear. Please see our answer to comment #23 on how we rephrased it.

35. Line 253: personnel cost is spent less.....

Answer: Amended as suggested.

36. Line 262-264: These two sentences seem to ignore an earlier sentence in which the cost for serotyping was included in the estimate (Line 258-260).

Answer: The reviewer is correct. We removed the statement on serotyping. It is now written as “However, WGS automatic bioinformatic pipelines can also provide other relevant information as part of their output, such as the presence of virulence or resistance genes”.

37. Line 264: remove “between”

38. Line 269: proven to be

Answer: Comment #37 and #38 amended as suggested

39. Line 291-294: This sentence needs to be re-written. The *sdg* gene is typically associated with Enteritidis and its absence casts a doubt on the serovar designation. The genetic distance from the other Enteritidis (line 288) also raises a concern about the designation.

Answer: We would like to refer to our answer to comments #15, which also answers this comment.

40. Line 398: Article number is not provided.

Answer: The page number was indeed missing. It is added in the revised manuscript.

41. Figure 2: Remove fractional numbers on the y axis since alleles are whole numbers.

Answer: Figure 2 has been updated so the y-axis only shows integers.

August 29, 2022

Dr. Roan Pijnacker
National Institute for Public Health and the Environment (RIVM), Centre for Infectious Disease Control
Bilthoven
Netherlands

Re: Spectrum01375-22R1 (Comparing multiple locus variable-number tandem repeat analyses with whole genome sequencing as typing method for Salmonella Enteritidis surveillance in the Netherlands, January 2019 to March 2020)

Dear Dr. Roan Pijnacker:

Your manuscript has been accepted, and I am forwarding it to the ASM Journals Department for publication. You will be notified when your proofs are ready to be viewed.

Sincerely,

Sadjia Bekal
Editor, Microbiology Spectrum

Journals Department
Review of the revised manuscript "Comparing multiple locus variable-number tandem repeat analyses with whole genome sequencing as typing method for *Salmonella* Enteritidis surveillance in the Netherlands, January 2019 to March 2020" (control no. Spectrum01375-22R1).

Summary of the major findings of the article:

This study demonstrates that WGS analysis provides higher resolution than MLVA typing for public health surveillance and outbreak detection of *Salmonella* Enteritidis, based on parallel typing of 439 isolates collected from human patients in the Netherlands over a 13 months period. The authors also show that there is a poor correlation between the clustering results obtained by the two subtyping methods.

Reviewer's overall impression of the paper:

The manuscript is well written, the methods are sound, the findings are presented in a clear and concise manner, and the conclusions are also supported by other studies done previously for different *Salmonella* serovars. This reviewer is also aware that similar results were obtained in a (still unpublished) study of Canadian isolates of *Salmonella* Enteritidis at the National Microbiology Laboratory in Canada (*personal communication*).

The findings presented in this study are very relevant for public health organizations, as the authors pointed out, for MLVA inaccuracy in outbreak delineation (compared to WGS) can hamper the source tracing efforts.

Comments on the revised manuscript:

The comments and suggestions from the reviewers have been adequately addressed by the authors in the revised manuscript.

Major shortcoming:

none

Comments:

- #1 Error in colors in the caption of Figure 5 (Lines 255-257): the legend refers to light blue and dark blue colors, but the bar chart actually appears to be rendered in blue and black. The second part of the caption should probably be changed to: "*Light blue = part of the outbreak based on cgMLST. **Black** = not part of the outbreak based on cgMLST.*"

#2 Minor typos should be fixed in the revised portion of the text (277165_1_art_file_6103250_rg1p64.pdf):

- Lines 244 and 310: typo, italicize S. in “*S.* Enteritidis”

- Line 246: typos: ... presumably got ill through ...” (instead of *presumable got ill though*)

-Line 349: Missing capital letters: “dr.” should be “Dr.” (“*We would like to thank Dr. Gerhard Falkenhorst and Dr. Sandra Simon...*”)